# A Comprehensive Analysis of Programmed Cell Death-Associated Genes for Tumor Microenvironment Evaluation Promotes Precise Immunotherapy in Patients with Lung Adenocarcinoma

**DOI:** 10.3390/jpm13030476

**Published:** 2023-03-06

**Authors:** Yunxi Huang, Wenhao Ouyang, Zehua Wang, Hong Huang, Qiyun Ou, Ruichong Lin, Yunfang Yu, Herui Yao

**Affiliations:** 1Guangdong Provincial Key Laboratory of Malignant Tumor Epigenetics and Gene Regulation, Department of Medical Oncology, Breast Tumor Center, Phase I Clinical Trial Center, Sun Yat-sen Memorial Hospital, Sun Yat-sen University, Guangzhou 510120, China; 2Department of Experimental Research, The Affiliated Tumor Hospital of Guangxi Medical University, Nanning 530000, China; 3Division of Science and Technology, Beijing Normal University-Hong Kong Baptist University United International College, Zhuhai 519000, China; 4School of Medicine, Guilin Medical University, Guilin 541000, China; 5Faculty of Medicine, Macau University of Science and Technology, Taipa, Macau 999078, China

**Keywords:** lung adenocarcinoma, programmed cell death, prognostic prediction, immunotherapy, tumor microenvironment

## Abstract

Immune checkpoint inhibitors (ICIs) represent a new hot spot in tumor therapy. Programmed cell death has an important role in the prognosis. We explore a programmed cell death gene prognostic model associated with survival and immunotherapy prediction via computational algorithms. Patient details were obtained from The Cancer Genome Atlas (TCGA) and Gene Expression Omnibus databases. We used LASSO algorithm and multiple-cox regression to establish a programmed cell death-associated gene prognostic model. Further, we explored whether this model could evaluate the sensitivity of patients to anti-PD-1/PD-L1. In total, 1342 patients were included. We constructed a programmed cell death model in TCGA cohorts, and the overall survival (OS) was significantly different between the high- and low-risk score groups (HR 2.70; 95% CI 1.94–3.75; *p* < 0.0001; 3-year OS AUC 0.71). Specifically, this model was associated with immunotherapy progression-free survival benefit in the validation cohort (HR 2.42; 95% CI 1.59–3.68; *p* = 0.015; 12-month AUC 0.87). We suggest that the programmed cell death model could provide guidance for immunotherapy in LUAD patients.

## 1. Introduction

Immune checkpoint inhibitors (ICIs) are a promising treatment for lung adenocarcinoma (LUAD) patients and have achieved considerable success in clinical usage. Yet, the response rate only reaches 30% of cancer patients [1]. Hence, it is urgent and indispensable to look for new biomarkers to identify more precisely the potential responders to ICI therapy.

Currently, tumor mutation burden (TMB), microsatellite instability-high (MSI-H), combined positive score (CPS), and tumor microenvironment (TME) are widely recognized markers for ICI responsiveness [2,3,4,5,6]. However, lack of an optimized integrative method to assess ICI responsiveness at earlier stages of treatment will inevitably lead to bias and poor efficiency. To our knowledge, immunotherapy response is closely related to prognosis for cancer patients, independently of tumor stage, pathological status, and radiomic characteristics [7].

Programmed cell death is a universal phenomenon in the development of organisms and is closely associated with tumorigenesis, to which apoptosis, ferroptosis, and autophagy are particularly relevant [8]. Several lines of evidence showed programmed cell death is an essential mechanism in tumorigenesis, by mediating the functional state of TME [9]. Tumor cells undergoing programmed cell death facilitate the recruitment of anticancer immune cells, thereby inducing granulocytes to kill the tumor cells, which further amplifies the immune response. However, the underlying mechanisms of this amplification loop are still unclear.

In this study, we constructed a systematic model based on programmed cell death-associated genes associated with survival using computational algorithms. We demonstrated that the programmed cell death signature could predict the immunotherapy sensitivity and prognosis of LUAD patients. Moreover, to explore the association between the programmed cell death signature and the characteristics of anti-tumor immunity, we analyzed in depth the functional enrichment of the predictive signature to determine the potential mechanisms of LUAD progression.

## 2. Materials and Methods

### 2.1. Patients and Study Design

This study was performed according to the Transparent Reporting of a Multivariable Prediction Model for Individual Prognosis or Diagnosis (TRIPOD) guideline. This study involving human participants was approved and reviewed by Sun Yat-sen Memorial Hospital, Sun Yat-sen University, Guangzhou, China. The exempt for informed consent of the participants was waived because the human data profiles were downloaded from public datasets. The workflow of our study is shown in Figure 1. The RNA-seq matrix and clinical data of 1342 LUAD patients (followed-up time for overall survival ≥1 month) were downloaded from The Cancer Genome Atlas (TCGA) portal (https://portal.gdc.cancer.gov/repository) and three datasets (GSE68465, GSE72094, and GSE135222) were downloaded from the Gene Expression Omnibus (GEO) database (https://www.ncbi.nlm.nih.gov/geo/, (accessed on 25 June 2022)). A total of 490 patients from TCGA were assigned to the training cohort, 386 patients from GSE72094 and 439 patients from GSE68465 were defined as validation cohort 1 and cohort 2. Moreover, GSE135222 as the validation cohort 3 with 27 non-small cell lung cancer patient treated with anti-PD-1/PD-L1.

Apoptosis-associated genes were obtained from the Molecular Signatures Database (http://www.gsea-msigdb.org/gsea/index.jsp, (accessed on 25 June 2022)). Autophagy-associated genes were obtained from the Human Autophagy Database (http://www.autophagy.lu/,(accessed on 25 June 2022)). Ferroptosis-associated genes were reviewed from earlier studies [10,11,12,13]. The mRNA matrices from TCGA, GSE68465, GSE72094, and GSE135222 were integrated and corrected in the “sva” R package, and the intersection genes were taken for further analysis. The primary end points were overall survival (OS) and progression-free survival (PFS).

### 2.2. Construction and Validation of the Programmed Cell Death Associated-Gene Models

The apoptosis-associated genes, ferroptosis-associated genes, and autophagy-associated genes differentially expressed were screened by the Wilcoxon test between tumor and adjacent normal tissues, with the selection criteria log (fold change) >1 for upregulated genes, and log (fold change) <−1 for downregulated genes. Univariate Cox regression analysis was used to identify the genes from the apoptosis, ferroptosis, and autophagy pathways associated with prognosis. Next, the LASSO Cox regression analysis and stepwise regression were applied to select essential genes to construct the prognostic model. Each patient’s risk score was calculated with the following formula:(1)$$Riskscore=\overset{n}{\underset{i=1}{\sum }}\mathrm{coef}\;i*\mathrm{expr}\;i$$
where coef is the coefficient, expr is the expression, coef i is the coefficient of gene i, and expr i is the expression of gene i. The patients were divided into low-risk and high-risk groups based on the optimum cut-off values of the apoptosis-, ferroptosis-, autophagy-, and programmed cell death-associated prognosis formula, which were given by the “surv_cutpoint” algorithm in the R package “survminer.” We used Kaplan–Meier survival curves to compare the OS of the different risk groups by the log-rank test. Time-dependent ROC analyses were used to evaluate the performance of each prognostic formula via R package “survivalROC.” In addition, we used the three validation cohorts to validate each prognostic formula.

For exploring whether programmed cell death risk score can be an independent predictor of lung adenocarcinoma, programmed cell death risk score model was combined with the clinical information from the TCGA database, including age, T stage, N stage. If the *p* value was less than 0.05 in the multivariate analysis, it was considered independent of prognostic factors.

### 2.3. Evaluation of the Tumor Microenvironment and Immunotherapy Response

We identified the enriched pathway in the high-risk and low-risk score groups using Gene Ontology (GO), and Kyoto Encyclopedia of Genes and Genomes (KEGG) used the R package “clusterProfiler” and set a threshold of significance of *p* < 0.05.

We assessed the differences in immune cells infiltration and immune pathways between the high-risk and low-risk score groups, as defined by the programmed cell death associated-gene model through single-sample gene set enrichment analysis (ssGSEA). Additionally, we calculated the estimated score, the stromal score, and the immune score of the TCGA cohort of patients using the “estimate” R package. We evaluated the tumor stem cell score by DNA stemness score and RNA stemness score (RNAss), which are based on DNA methylation pattern (DNAss) and mRNA expression, respectively [14].

Additionally, we scored the tumor immunogenicity which was at different levels using the Cancer Immunome Atlas (https://tcia.at/, (accessed on 25 June 2022)), and referred to as immunophenoscore (IPS) [15]. IPS were used to evaluate the immunotherapy response in LUAD patients. Furthermore, we used the validation cohort 3 from patients with non-small cell lung cancer treated by anti-PD-1/PD-L1 to validate the immunotherapy response.

### 2.4. Cell Culture

The normal human normal bronchial epithelium cell line (BEAS-2B) and human non-small cell lung cancer cell line (HCC827) were cultured in DMEM medium with 10% fetal bovine serum (FBS) and 1% penicillin–streptomycin. The other human non-small cell lung cancer cell line (A549) was cultured in RPMI-1640 medium with 10% fetal bovine serum (FBS) and 1% penicillin–streptomycin. All cells were incubated in a humidified atmosphere containing 5% CO_2_ at 37 °C.

### 2.5. RNA Isolation, cDNA Synthesis, and qPCR Analysis

TRIzol reagent (Thermo Fisher Scientific, Inc., Boston, MA, USA) was used to isolate total RNA from BEAS-2B, HCC827, and A549 and were further reverse transcribed to cDNA using the HiScript III first Strand cDNA Synthesis Kit (Vazyme Biotech Co., Ltd., Nanjing, China) according to the manufacturer’s instructions. Furthermore, we used the ChamQ Universal SYBR qPCR Master Mix kit available from Vazyme (Vazyme Biotech Co., Ltd., Nanjing, China) as well as the Quant Studio TMDx from Applied Biosystems. The polymerase was activated at 95 °C for 30 s, then cycled for 40 cycles at 95 °C for 5 s and 60 °C for 30 s. Using the relative quantification 2^−ΔΔCT^ method, the relative level was calculated. Appendix A contains a list of all primer sequences.

### 2.6. Statistical Analysis

We determined the genes differentially expressed in tumors and adjacent normal tissues using the Wilcoxon test. Kaplan–Meier survival analysis between different groups was performed by the log-rank test. We used the R package “survminer” to calculate the optimal cutoff value for continuous variables [15]. The area under the ROC curve (AUC) was used to evaluate the prognostic performance of the model via R package “survivalROC.” All statistical analyses were performed in the R software (Version 4.0.0). *p* < 0.05 was considered statistically significant.

## 3. Results

In total, 1342 patients were included. A total of 490 LUAD patients from the TCGA cohort (228 [46.53%] male, mean [SD] age, 65 [10.01]), 386 patients from the validation cohort 1 (168 [43.52%] male, mean [SD] age, 69 [9.41]), and 439 patients from validation cohort 2 (221 [50.34%] male, mean [SD] age, 64 [10.08]), and 27 non-small cell lung cancer patients from the immunotherapy validation cohort 3 (22 [81.2%] male, mean [SD] age, 62 [8.95]) were included. It summarizes the detailed clinical information of the enrolled patients in Appendix A. PCA results show that the data of four queues are clearly separated before batch correction (Appendix A), and the data of each queue are evenly distributed after batch correction (Appendix A).

### 3.1. Identification of Apoptosis-, Ferroptosis-, and Autophagy-Associated Differentially Expressed Genes

Among the 490 patients from the TCGA cohort, a total of 38 apoptosis-associated differentially expressed genes (DEGs) (17 upregulated, 21 downregulated) were identified by comparison between tumors and adjacent normal tissues (Appendix A). These DEGs were used to perform GO and KEGG pathway enrichment analyses to elucidate the biological functions and pathways involved in apoptosis. We found that the apoptosis-associated DEGs were enriched in apoptosis, C-type lectin receptor signaling pathway, regulation of apoptotic signaling pathway, muscle organ development, and extrinsic apoptotic signaling pathway (Appendix A).

In total, 15 ferroptosis-associated DEGs (10 upregulated, 5 downregulated) were identified by comparison between tumors and adjacent normal tissues (Appendix A). GO and KEGG enrichment were used to analyze the biological functions and pathways supported by the ferroptosis-associated DEGs. Ferroptosis-associated DEGs were enriched in functions related to response to oxidative stress, cellular response to oxidative stress, cellular response to chemical stress, and ferroptosis, cysteine, and methionine metabolism (Appendix A).

Finally, a total of 29 autophagy-associated DEGs (15 upregulated, 14 downregulated) were identified by comparison between tumors and adjacent normal tissues (Appendix A). Using the same strategy, we found that autophagy-associated DEGs were enriched in functions related to regulation of autophagy, intrinsic apoptotic signaling pathway, IL-17 signaling pathway, and mitophagy (Appendix A).

### 3.2. Apoptosis-Associated Gene Risk Score Associated with Survival

Using univariable analysis, we found that nine of the apoptosis-associated genes were associated with the OS of the LUAD patients. To prevent the model from over-fitting, we further used LASSO regression analysis (Appendix A) and selected six genes to construct an apoptosis signature formula as follows: apoptosis risk score = (F2 × 0.105) − (GPX3 × 0.176) + (BCL2L10 × 0.188) + (PAK1 × 0.14) + (KRT18 × 0.217) − (BTG2 × 0.160). This formula was used to calculate the risk score of each patient. Those patients were divided into a high-risk score group (*n* = 293) or a low-risk score group (*n* = 197) based on an optimal cut-off value. Consistently, the Kaplan–Meier curve demonstrated that the patients in the high-risk score group had considerably worse OS than those in the low-risk score group (HR 2.34, 95% CI 1.67–3.27, *p* < 0.001) (Appendix A). The performance of the apoptosis formula was evaluated by time-dependent ROC curves with the AUC reaching 0.67 at 1 year, 0.70 at 3 years, and 0.67 at 5 years (Appendix A).

To test the stableness of the apoptosis model constructed from the TCGA cohort, we calculated the individual apoptosis risk scores of the patients from the validation cohort 1 and validation cohort 2. The patients from the two validation cohorts were classified into the high and low apoptosis risk score groups using the optimal cut-off score calculated with the same formula as that used for the training cohort. Patients with a high apoptosis risk score were more likely to have worse OS compared to patients with a low apoptosis risk score in both cohort 1 (HR 1.94, 95% CI 1.28–2.94, *p* = 0.0016) (Appendix A) and cohort 2 (HR 1.39, 95% CI 1.06–1.83, *p* = 0.016) (Appendix A). Moreover, the apoptosis signature risk score appeared as a powerful predictive value in cohort 1, with AUC of 0.67, 0.64, and 0.61 (Appendix A), and in cohort 2, with AUC of 0.63, 0.59, and 0.59 (Appendix A).

### 3.3. Ferroptosis-Associated Gene Risk Score Associated with Survival

We used univariable and stepwise regression analyses to screen out two ferroptosis genes (Appendix A), which were used to construct the following prognosis formula: ferroptosis risk score = (SLC7A11 × 0.084) + (GCLC × 0.093). This formula was used to calculate the ferroptosis risk scores of all patients and classified these patients into the high (*n* = 52) or low ferroptosis risk score group (*n* = 438), based on the optimal cut-off value. Consistently, the Kaplan–Meier curve demonstrated that the patients in the high ferroptosis risk score group had considerably worse OS than those in the low ferroptosis risk score group (HR 2.12, 95% CI 1.43–3.14, *p* < 0.001) (Appendix A) with 1-, 3-, and 5-year AUC reaching 0.60, 0.59, and 0.53, respectively (Appendix A).

We verified the stableness of the ferroptosis-associated formula using cohorts 1 and 2. Similarly, the patients from these two cohorts were divided into high and low ferroptosis risk score groups using the optimal cut-off of the ferroptosis risk score calculated with the same formula as for the training cohort. Patients in the high ferroptosis risk score group were more likely to die earlier compared with those from the low ferroptosis risk score group in both cohort 1 (HR 1.81, 95% CI 1.03–3.17, *p* = 0.037) (Appendix A) and cohort 2 (HR 1.48, 95% CI 1.00–2.17, *p* = 0.046) (Appendix A). Moreover, the ferroptosis gene signature risk score formula calculated a powerful predictive value in cohort 1, with AUC of 0.61, 0.62, and 0.66 (Appendix A), and cohort 2, with AUC of 0.58, 0.60, and 0.58 (Appendix A).

### 3.4. Autophagy-Associated Gene Risk Score Associated with Survival

In our univariable analysis, seven of the apoptosis-associated genes were related to OS in LUAD patients. To prevent the model from over-fitting, we further used LASSO Cox regression analysis (Appendix A) and shortlisted three genes to construct the autophagy signature formula, as follows: autophagy risk score = (ITGB4 × 0.140) + (GAPDH × 0.0.514) + (BIRC5 × 0.111). This formula was used to calculate the autophagy risk scores of all patients, who were stratified into the high autophagy risk score group (*n* = 110) or the low autophagy risk score group (*n* = 380), based on the optimal cut-off value. Consistently, the Kaplan–Meier curve demonstrated that the patients in the high-risk group had considerably worse OS than those in the low-risk score group (HR 2.59, 95% CI 1.90–5.53, *p* < 0.0001) (Appendix A). Time-dependent ROC curves were generated to test the survival prediction value of the prognostic model and indicated AUC of 0.68 at 1 year, 0.69 at 3 years, and 0.60 at 5 years (Appendix A).

In order to test the stableness of the model constructed from the TCGA cohort, we calculated the individual autophagy signature risk scores of the patients from validation cohort 1 and validation cohort 2. The patients from the two cohorts were also divided into high and low autophagy risk score groups based on the optimal value calculated with the same formula as for the training cohort. Similarly, patients in the high autophagy risk score group were more likely to die earlier compared with those in the higher autophagy risk score group in both cohort 1 (HR 2.41, 95% CI 1.63–3.57, *p* < 0.0001) (Appendix A) and cohort 2 (HR 1.62, 95% CI 1.22–2.16, *p* < 0.001) (Appendix A). Moreover, the apoptosis signature risk score proved to be of powerful predictive value in validation cohort 1, with AUC of 0.67, 0.68, and 0.74 (Appendix A), and cohort 2, with AUC of 0.69, 0.66, and 0.61 (Appendix A).

### 3.5. Programmed Cell Death-Associated Gene Risk Score Associated with Survival

Considering a potential crosstalk between different types of programmed cell death in the TME, we integrated the apoptosis-associated gene model, the ferroptosis-associated gene model, and the autophagy-associated gene model to generate the programmed cell death-associated gene model. We found that patients with a high programmed cell death risk score had a higher probability of earlier death than those with a low programmed cell death risk score. The programmed cell death signature formula reads as follows: programmed cell death risk score = (F2 × 0.113) − (GPX3 × 0.178) + (BCL2L10 × 0.143) + (KRT18 × 0.079) + (GCLC × 0.007) − (BTG2 × 0.06) + (SLC7A11 × 0.013) + (ITGB4 × 0.113) + (GAPDH × 0.406) − (BIRC5 × 0.010). This formula was used to calculate the risk scores of the patients. The patients were accordingly stratified into a high programmed cell death risk score group (*n* = 52) or a low programmed cell death risk score group (*n* = 438) based on the optimal cut-off value (Figure 2A–C). Then, the Kaplan–Meier curve demonstrated that the patients with high programmed cell death risk score had considerably worse OS than those with a low risk score (HR 2.70, 95% CI 1.94–3.75, *p* < 0.0001) (Figure 2D). Time-dependent ROC curves were used to evaluate the prediction performance of the model and showed 1-, 3-, and 5-year AUC of 0.71, 0.71, and 0.66, respectively (Figure 2E). These results demonstrated that the integrated programmed cell death-associated gene model was better than the component models for the prediction of patient prognosis.

Similarly, we calculated the individual programmed cell death signature risk scores of the patients from validation cohort 1 and validation cohort 2. Accordingly, the patients from the two cohorts were divided into high-risk or low-risk score groups using an optimal value calculated with the same formula as for the training cohort (Figure 3A–C and Figure 4A–C). Similarly, patients in the high risk score group were more likely to die earlier compared with patients in the higher risk score group in both cohort 1 (HR 2.42, 95% CI 1.59–3.68, *p* < 0.0001) (Figure 3D) and cohort 2 (HR 1.49, 95% CI 1.14–1.94, *p* = 0.0032) (Figure 4D). Moreover, the programmed cell death signature risk score served as a powerful predictive value in cohort 1, with 1-, 3-, and 5-year AUC of 0.68, 0.69, and 0.71, respectively (Figure 3E), and cohort 2, with 1-, 3-, and 5-year AUC of 0.66, 0.63, and 0.60, respectively (Figure 4E). Additionally, co-expression networks results showed that the expression of programmed death genes is mainly positively correlated, which indicated that these genes have synergistic effects (Appendix A).

### 3.6. Programmed Cell Death-Associated Gene Risk Score Associated with Tumor Immune Microenvironment

Next, we sought to clarify the biological functions and pathways related to the programmed cell death model. The DEGs (Appendix A) associated with the integrated programmed cell death model were enriched mainly in functions linked to cell division, including cell cycle, chromosome separation, organelle division, nuclear chromosome segregation, and sister chromatid separation (Appendix A). Moreover, through multivariate cox regression analyses, the DEGs, such as TIMP1, CAV1, GAPDH, IER3, BCL2L10, BTG2, F2, GPX3 are regarded as the independent factors for the prognosis of LUAD (Appendix A).

Furthermore, we analyzed the correlation between the programmed cell death risk score and immune cells and pathways. We quantified the enrichment fractions of different immune cell subsets, and the associated functions or pathways with ssGSEA. Interestingly, in the TCGA cohort, the risk scores of dendritic cells, B cells, CD8+ T cells, helper T cells, mast cells, neutrophils, and tumor-infiltrating lymphocytes were higher in the low-risk score group than in the high-risk score group (*p* < 0.05) (Figure 5A). Moreover, the level of immune checkpoint molecules, HLA, T-cell costimulatory pathway, and type II interferon response was higher in the low-risk score group than in the high risk score group, whereas the level of type I MHC was just the opposite (*p* < 0.05) (Figure 5A). We found that compared with the high risk score group, the low risk score group had a higher estimate score, immune score, and stromal score (Figure 5B,C). Moreover, the programmed cell death risk score was positively correlated with tumor stem cell (Figure 5E,F), which demonstrated that patients with a high risk score may be more likely to experience relapse.

### 3.7. Programmed Cell Death-Associated Gene Risk Score Associated with Immunotherapy Response

IPS is a scoring feature of machine learning that can predict patients’ response to ICI therapy. We observed that the risk score of programmed cell death-associated model was inversely correlated with TME. Therefore, we further explored whether programmed cell death could affect the level of immune checkpoint molecule expression. For this purpose, we used two subtypes of IPS values (IPS-CTLA-4 and IPS-PD-1/PD-L1) as surrogates for the measurement of LUAD patients’ responses to anti-CTLA-4 and anti-PD-1/PD-L1 treatments (Figure 6A,B). In the corresponding prognostic formula, the relative probability of the low-risk score group to respond to anti-CTLA-4 and anti-PD-1/PD-L1 treatments was higher than that in the high-risk score group, which indicates that patients with a low-risk score might be suitable for ICI treatment. Moreover, we further verified our results on the immunotherapy cohort (validation cohort 3, GSE135222), which yielded a HR of 2.42 (95% CI 1.59–3.68, *p* = 0.015) (Figure 6C), and a 12-month predicted AUC of 0.87 (Figure 6D). These results demonstrated that the patients with low programmed cell death risk scores who received anti-PD-1/PD-L1 treatment had better prognosis of progression-free survival than the those with a high programmed cell death risk score.

By univariate analysis, except for age (*p* = 0.847), tumor T stage (*p* < 0.001), M stage (*p* = 0.026), N stage (*p* < 0.001), and programmed cell death risk score (*p* < 0.001) can be regarded as prognostic factors of lung adenocarcinoma, but in multivariate analysis, only T stage (*p* = 0.003) and programmed cell death risk score (*p* < 0.001) are regarded as independent prognostic factors of lung adenocarcinoma(Figure 7A,B). We further analyzed subgroups of T staging and found that patients with high programmed cell death risk score had shorter OS, regardless of early T staging (*p* < 0.001) or late T staging (*p* = 0.011) (Figure 7C,D). To verify programmed cell death risk score as a predictor of immunotherapy, we found that patients with low programmed cell death risk score high level of PD-1 (*p* < 0.05), PD-L1 (*p* < 0.05), CTLA4 (*p* < 0.001), TIM-3 (*p* < 0.05), LAG3 (*p* > 0.05) (Figure 8A–E). Then, multivariate ROC showed that programmed cell death risk score (AUC:0.705) was more effective than PD-1 (AUC:0.453), PD-L1(AUC:0.497), CTLA4(AUC:0.417), TMB(AUC:0.506), LAG3(AUC:0.485), TIM-3(AUC:0.419), and MSI(AUC:0.475) in predicting the prognosis of lung adenocarcinoma (Figure 8F). We found that patients with low programmed cell death risk score high level of plasma cells, CD4 memory resting T cells, monocytes, dendritic cells resting, mast cells resting, and mast cells activated (Figure 8G).

Validation of the expression level of the genes from prognostic model in the cell lines. We used three cell lines BEAS-2B, A549, and HCC827 to verify the expression of genes associated with the programmed cell death-related prognostic model. The results showed that the expression of BCL2L10, BIRC5, F2, KRT18, and ITGB4 significantly decreased, whereas expression of BTG2 and GPX3 significantly increased in the A549 and HCC827 group compared with the control group. Moreover, expression of GAPDH and GCLC significantly decreased in the A549 group compared with BEAS-2B. Expression of SLC7A11 significantly decreased in the HCC827 group compared with BEAS-2B (Figure 9A–J).

## 4. Discussion

In this study, we explored the role of programmed cell death-associated genes in predicting the therapeutic effect of ICIs in LUAD patients. We found that nine apoptosis-associated gene signatures, two ferroptosis-associated gene signatures, and seven autophagy-associated gene signatures were related to the survival rate of LUAD patients. By combining all these cell death-associated gene signatures, we constructed a programmed cell death-associated predictive gene model. We found that this model was strongly associated with immune cells infiltration. our findings are similar to the previous research showing that the tumor immune microenvironment has an influence on cancer cell death. Surprisingly, compared with the high-risk group, the low-risk group had more immune cell infiltration and better response to ICI therapy.

Recently, immune checkpoint blockade has become the most promising therapy for cancer treatment. Blocking immune checkpoint molecules, which allows T cells to proliferate and kill the tumor cells, has achieved considerable success in clinical treatment for LUAD patients, especially the use of PD-1 and CTLA-4 inhibitors [16]. Multiple clinical trials demonstrated that combined therapy with anti-PD-1/PD-L1 and anti-CTLA-4 was more effective than monotherapy [17,18]. However, not all patients could benefit from a combined therapy, which is also associated with increased side effects. Therefore, patient selection is crucial and must be done before choosing a treatment plan. In this study, we found that the programmed cell death formula was effective to predict patient’s sensitivity to anti-PD-1/PD-L1 and anti-CTLA4 treatments and may be beneficial to the treatment of LUAD.

Recently, with the deepening of our understanding of programmed cell death, many researchers noticed the role of this biological process in the TME. Cell death may cause cell swelling and rupture of the plasma membrane, which increases the immunogenicity of dying cells. The transformation of dying cells from non-immunogenic to immunogenic cells, termed immunogenic cell death (ICD), has the potential to increase the percentage of cytotoxic T lymphocyte (CTLs) infiltration. The activation of CD8+ T cells can further suppress tumors via the induction of programmed cell death. However, the exact mechanisms underlying the effect of programmed cell death on TME remain unclear. Therefore, we evaluated the immune cells and functions in LUAD patients using an ssGSEA algorithm, which demonstrated that higher numbers of CD8+ T cells, Th1 cells, and Th2 cells, dendritic cells, as well as higher checkpoint molecule expression, were positively correlated with low programmed cell death risk scores.

A lot of evidence suggests that different cell deaths could coexist in the TME and interfere with each other [19,20,21]. Most previous studies, however, have focused on a single factor in the TME, with little attention paid to the role of programmed cell death as a whole. Unlike previous studies, we first looked at programmed cell death as a whole to predict TME, which may increase the accuracy of the model and reduce bias. Hence, given the specific role of programmed cell death in TME, a combined programmed cell death-associated model predicting the efficacy of immunotherapy may bring a novel perspective for LUAD treatment. Moreover, the programmed cell death-associated model helps researchers to deeply understand the correlation between TME and programmed cell death. For instance, the low programmed cell death risk score group contained more stromal cells and immune cells than the high programmed cell death risk score group.

Through our previous study, we found that MUC16 mutation status and protein tyrosine phosphatase receptor T mutation were reliably predictive of immunotherapy outcome [22,23]. Subsequently, we proposed to integrate a new lncRNA score with TMB, CTL infiltration, and PD-L1 expression to predict immunotherapy responses [24]. To our knowledge, lncRNA was the essential regulator of mRNA expression in tumor progression. In comparison, our present study constructed a programmed cell death-associated mRNA expression signature model based on apoptosis, ferroptosis, and autophagy-associated genes. This model can be regarded as an immunotherapeutic biomarker that can increase and improve substantially the ability to predict immunotherapeutic benefit, likely because of the additional information provided by the programmed cell death-associated mRNA expression, which has the potential to integrate multiple parameters and reduce bias. Furthermore, this model is the first study to integrate factor of programmed cell death for predicting the therapeutic effect of ICIs, and may have the ability to maximize ICI effects.

In addition, in our prior clinical trial [25], we used multi-omics approaches, including metabolism, radiomics, clinic-pathological characteristics, and genetics to predict the efficacy of neoadjuvant immunotherapy combined with chemotherapy response in non-small cell lung cancer patients (*n* = 40). In this study, we subsequently constructed a satisfactory model to predict the OS and immunotherapy response of patients with LUAD. In the near future, we will use our programmed cell death model to validate our clinical trial data and to investigate the potential mechanism of immunotherapy response.

Although our results are promising, there are still several limitations. First, the clinical situations of the patients are complex, and several variables may affect patient prognosis. Hence, it will be essential to verify the signature in large-scale multicenter cohorts. Second, the underlying crosstalk mechanism between programmed cell death-associated genes and immune environment in LUAD remains poorly understood and warrants further investigation. Finally, different types of cell death may occur simultaneously in tumors. Therefore, exploring other unusual types of cell death may help us obtain a comprehensive understanding of the role of cell death in tumorigenesis.

In summary, we constructed a programmed cell death-associated gene prediction model. We found that programmed cell death may affect TME and that anti-tumor-associated immune cells were more enriched in the low-risk score group. Additionally, the prediction of patients’ sensitivity to ICIs is conducive to precision medicine for LUAD patients. In the near future, we expect that research on programmed cell death will considerably improve the survival rate of cancer patients.

## Figures and Tables

**Figure 1 jpm-13-00476-f001:**
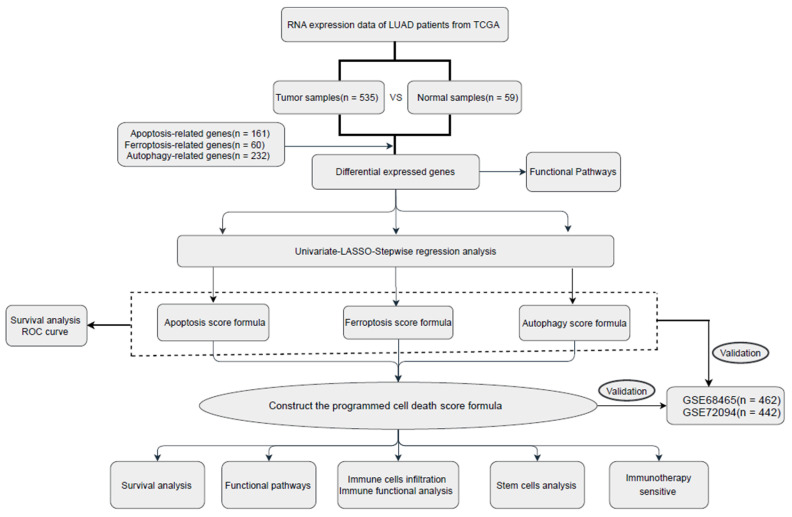
Workflow of this study.

**Figure 2 jpm-13-00476-f002:**
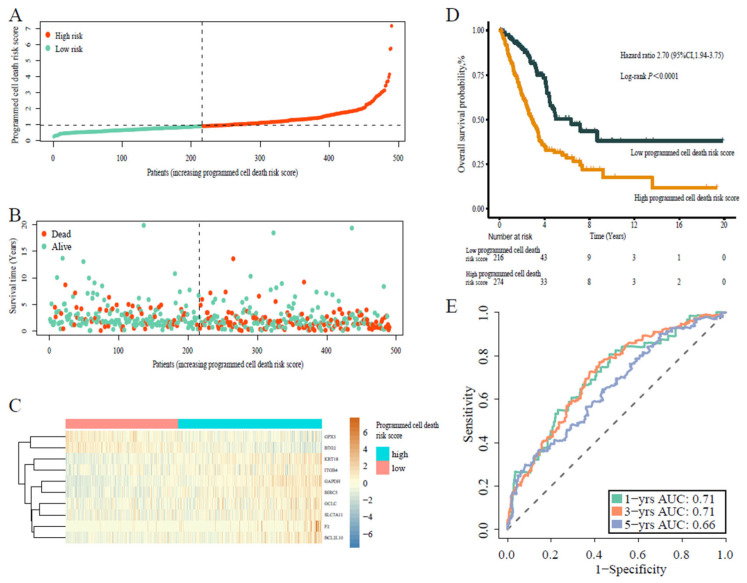
Prognostic analysis of programmed cell death risk score formula in the training cohort. (**A**) Distribution of the programmed cell death risk scores in the training cohort. (**B**) Patterns of the survival time and survival status of the patients based on programmed cell death risk scores. (**C**) Heatmaps of the ten prognostic genes for each patient of the training cohort. (**D**) Kaplan–Meier survival curve of the patients based on programmed cell death risk scores. (**E**) Time-related ROC analysis showing the prognostic performance of the programmed cell death risk score formula in the training cohort.

**Figure 3 jpm-13-00476-f003:**
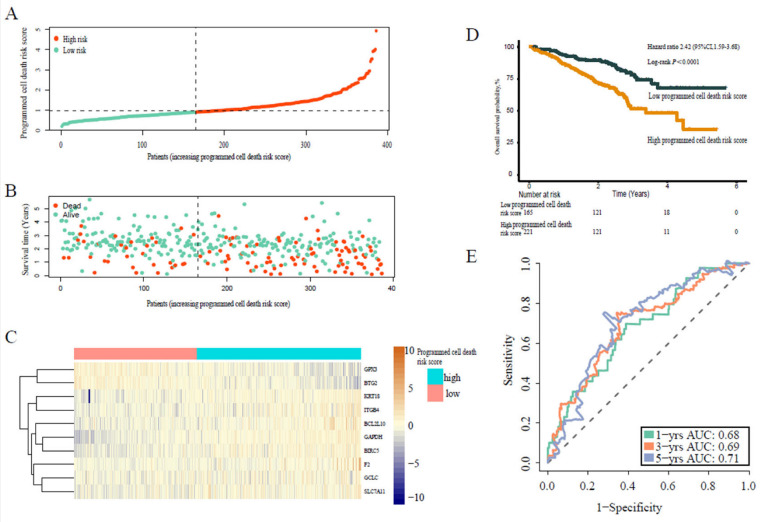
Prognostic analysis of programmed cell death risk score in validation cohort 1. (**A**) Distribution of programmed cell death risk scores in validation cohort 1. (**B**) Patterns of the survival time and survival status of the patients based on programmed cell death risk scores. (**C**) Heatmaps of the ten prognostic genes for each patient of the training cohort. (**D**) Kaplan–Meier survival curve of the patients according to the programmed cell death risk scores. (**E**) Time-related ROC analysis showing the prognostic performance of the programmed cell death risk score formula in validation cohort 1.

**Figure 4 jpm-13-00476-f004:**
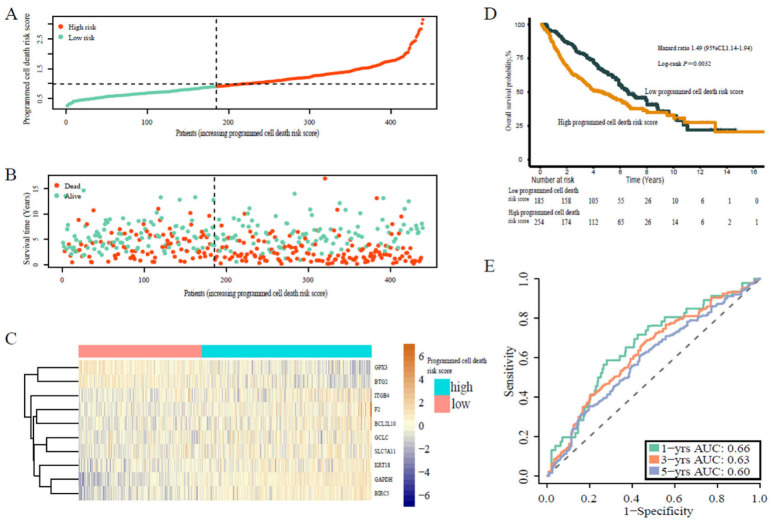
Prognostic analysis of programmed cell death risk score in validation cohort 2. (**A**) Distribution of programmed cell death risk scores in validation cohort 2. (**B**) Patterns of the survival time and survival status of the patients based on the programmed cell death risk scores. (**C**) Heatmaps of the ten prognostic genes for each patient of the training cohort. (**D**) Kaplan–Meier survival curve of the patients based on the programmed cell death risk scores. (**E**) Time-related ROC analysis showing the prognostic performance of the programmed cell death risk score formula in validation cohort 2.

**Figure 5 jpm-13-00476-f005:**
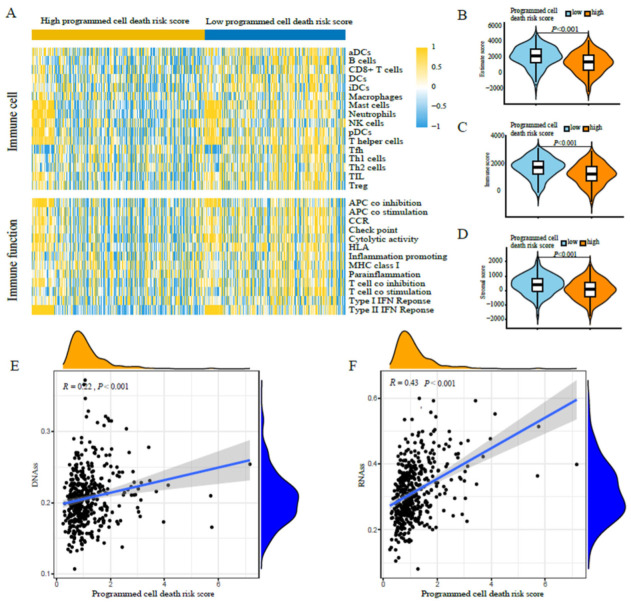
Exploration of differences in tumor immune microenvironment between the high and low programmed cell death risk score groups of the training cohort. (**A**) ssGSEA for the association between immune cell subpopulations and related functions. (**B**) The relationship between programmed cell death risk score and the estimate risk score. (**C**) The relationship between programmed cell death risk score and the immune risk score. (**D**) The relationship between programmed cell death risk score and the stromal risk score. (**E**) The programmed cell death risk score was positively correlated with DNAss. (**F**) The programmed cell death risk score was positively correlated with RNAss.

**Figure 6 jpm-13-00476-f006:**
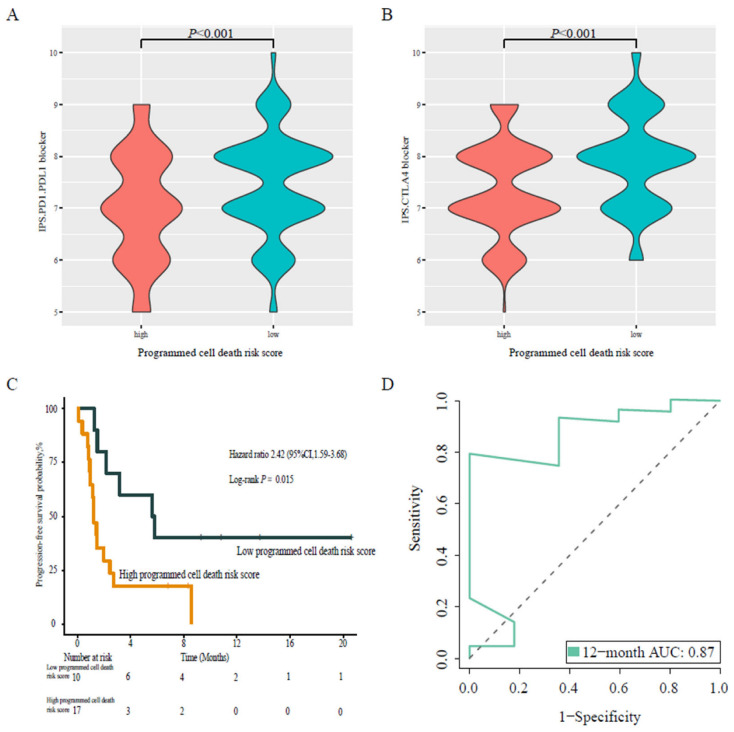
Validation of the programmed cell death risk score in the immunotherapy cohort. (**A**) Violin plots showing the difference of PD-1/PD-L1 IPS between the high and low programmed cell death risk score groups. (**B**) Violin plots showing the difference of CTLA4 IPS between the high and low programmed cell death risk score groups. (**C**) Kaplan–Meier survival curve of non-small cell lung cancer patients who were treated with anti-PD-1/PD-L1. (**D**) ROC analysis showing the immunotherapy prognostic performance of the programmed cell death risk score formula.

**Figure 7 jpm-13-00476-f007:**
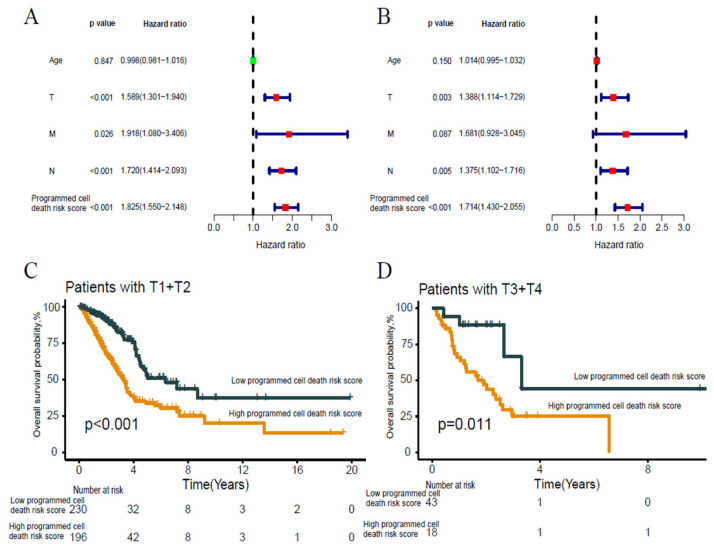
Analysis of independent prognostic factors in lung adenocarcinoma. (**A**) Univariate cox regression analyses for the clinical characteristics. (**B**) Multivariate cox regression analyses for the clinical characteristics. (**C**) KM survival curve of programmed cell death risk score in patients with T1 or T2. (**D**) KM survival curve of programmed cell death risk score in patients with T3 or T4.

**Figure 8 jpm-13-00476-f008:**
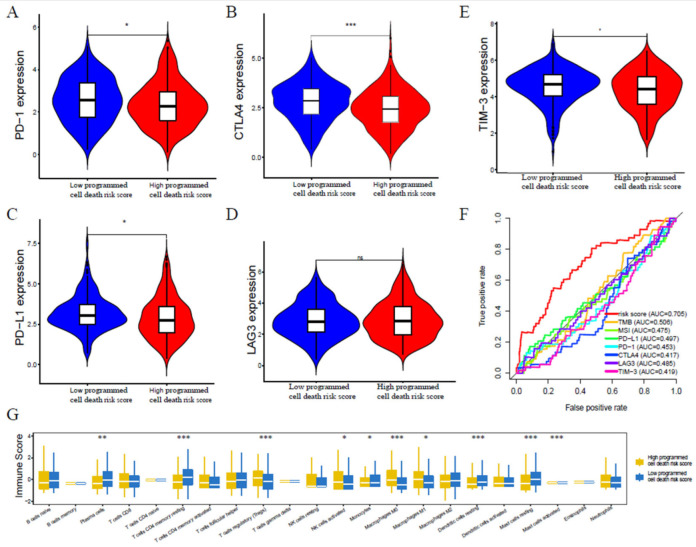
Programmed cell death risk score as the predictor for immunotherapy. (**A**) PD-1 expression difference between high and low programmed cell death risk score patients. (**B**) CTLA4 expression difference between high and low programmed cell death risk score patients. (**C**) PD-L1 expression difference between high and low programmed cell death risk score patients. (**D**) LAG3 expression difference between high and low programmed cell death risk score patients. (**E**) TIM3 expression difference between high and low programmed cell death risk score patients. (**F**) Multivariate ROC of programmed cell death risk score, PD-1, PD-L1, CTLA4, LAG3, TIM3, TMB and MSI. (**G**) Immune cells infiltration difference between high and low Programmed cell death risk score patients. ∗ *p* < 0.05, ∗∗ *p* < 0.01, ∗∗∗ *p* < 0.001.

**Figure 9 jpm-13-00476-f009:**
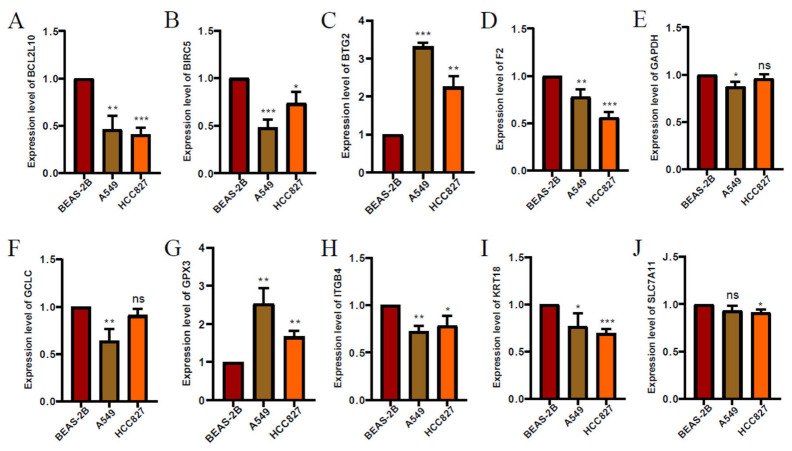
The expression levels of each gene from prognostic model. (**A**) (BCL2L10), (**B**) (BIRC5), (**C**) (BTG2), (**D**) (F2), (**E**) (GAPDH), (**F**) (GCLC), (**G**) (GPX3), (**H**) (ITGB4), (**I**) (KRT18), (**J**) (SLC7A11) in lung adenocarcinoma cell lines detected by RT-qPCR. ∗ *p* < 0.05, ∗∗ *p* < 0.01, ∗∗∗ *p* < 0.001.

## Data Availability

The RNA-seq matrix and clinical data of LUAD patients can be downloaded from The Cancer Genome Atlas (TCGA) portal (https://portal.gdc.cancer.gov/repository, (accessed on 25 June 2022)) and the Gene Expression Omnibus (GEO) database (https://www.ncbi.nlm.nih.gov/geo/, (accessed on 25 June 2022)).

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
