# Peer review of "A Comprehensive Analysis of Programmed Cell Death-Associated Genes for Tumor Microenvironment Evaluation Promotes Precise Immunotherapy in Patients with Lung Adenocarcinoma"

_jpm, 2023, doi:10.3390/jpm13030476_

Round 1

Reviewer 1 Report

The review presented here analyzes the relationship between programmed cell death genes and the tumor microenvironment. Despite the interesting content of the manuscript, which includes consideration of genes associated with ferroptosis and autophagy, there are a number of inaccuracies that need to be corrected:

1. Clinical characteristics of the patients are not presented

2. the genes associated with pyroptosis can also be considered

3. Should not be limited to PD-1/PD-L1 axis and CTLA-4, but also include other genes like LAG3, TIM-3.

4. The immune microenvironment, which includes different cell types, should be differentiated.

Author Response

Response to Reviewer 1

The review presented here analyzes the relationship between programmed cell death genes and the tumor microenvironment. Despite the interesting content of the manuscript, which includes consideration of genes associated with ferroptosis and autophagy, there are a number of inaccuracies that need to be corrected:

  1. Clinical characteristics of the patients are not presented.

Response: Thanks for your suggestion. The clinical characteristics of the patients were presented in supplementary Table 1.

  1. the genes associated with pyroptosis can also be considered

Response: Thank you for your great advice. We attempted to analyze pyroptosis related genes, but we found that these genes did not significantly improve the AUC in the model and the validation failed. Therefore, we did not try to add the pyroptosis-related prognostic model into the programmed death model. But we still think your suggestion is very good, because pyroptosis is a new way of programmed cell death.

  1. Should not be limited to PD-1/PD-L1 axis and CTLA-4, but also include other genes like LAG3, TIM-3.

Response: Thank you for your great advice. Currently, the drugs on the market are mainly anti-PD-1 /PD-L1 and anti-CTLA4, so we originally intended to use this model to predict the sensitivity of anti-PD-1 /PD-L1 and anti-CTLA4 treatment. However, LAG3, TIM-3, these new targeted drugs of immune checkpoint are expected to be widely used in the future. Therefore, Based on your suggestions, we feel it is necessary to analyze the correlation between other immune checkpoints and programmed death score to improve the application effectiveness of this model(Figure8 A-F).

4.The immune microenvironment, which includes different cell types, should be differentiated.

Response: Thank you for your great advice. We used CIBERSORT algorithm to annotate 22 kinds of immune cells and found that there was significant difference in immune cell infiltration between the two groups of patients in the prognostic model.

Reviewer 2 Report

The present study reports on dataset correlations between different programmed cell death programs and tumorigenesis, including overall survival, prognosis, etc. 

The authors introduce that cell death can drive immunogenicity and hence anti-tumor immunity but then show that low cell death risk is associated with improved survival. The authors claim that indeed cell death is important for anti-tumor immunity. 

There are two potential issues: either the text and figures are misleading or the interpretation of the data and conclusions are just wrong. 

To me it appears that low cell death risk correlates with improved survival and therefore the overall conclusion is that the fewer cells die, better for your cancer. This is indeed counterintuitive, but data is data... This is also the opposite to what the authors conclude.

Perhaps I don't understand the paper. This means that the paper needs to be entirely revisited and re-written so it can be understandable. Alternatively, if the figures show what I think they show, the conclusions need to be changed to go in line with their findings and not with the initial hypothesis. 

Author Response

Response to Reviewer 2

The present study reports on dataset correlations between different programmed cell death programs and tumorigenesis, including overall survival, prognosis, etc.

The authors introduce that cell death can drive immunogenicity and hence anti-tumor immunity but then show that low cell death risk is associated with improved survival. The authors claim that indeed cell death is important for anti-tumor immunity.

There are two potential issues: either the text and figures are misleading or the interpretation of the data and conclusions are just wrong.

To me it appears that low cell death risk correlates with improved survival and therefore the overall conclusion is that the fewer cells die, better for your cancer. This is indeed counterintuitive, but data is data... This is also the opposite to what the authors conclude.

Perhaps I don't understand the paper. This means that the paper needs to be entirely revisited and re-written so it can be understandable. Alternatively, if the figures show what I think they show, the conclusions need to be changed to go in line with their findings and not with the initial hypothesis.

Response: Thank you for your comments. In our study, we mainly included apoptosis, ferroptosis and autophagy-related genes to construct programmed death risk score, which was mainly based on cox risk model. Therefore, according to the influence of genes on survival of lung adenocarcinoma patients, different coefficients were assigned to genes. Genes with positive coefficients may inhibit programmed death, while those with negative coefficients mainly promote programmed death. For example, BCL2L10, whose cox regression coefficient is +0.143, has been reported to inhibit apoptosis. Sorry for our naming of the risk score which is confusing to the reader, and we will describe our prognostic model in a better way.

Reviewer 3 Report

The manuscript by Huang et al. sounds interesting at first glance, but I cannot complete my review at this time as the supplementary information is not included in the submitted version. Please find my comments below that I collected during my first read-through until mention of the missing supplementary figures (Line 176 on page 6). Please re-submit the manuscript with the supplementary figures, and I am happy to re-review.

Major comments

1.       While readable, please consider having a (near) native English speaking colleague or a professional editing service edit the manuscript. This will improve readability greatly.

2.        

Minor comments

General

Why isn’t programmed cell death abbreviated to PD-1?

It would be great if a little bit more mechanistic introduction could be provided – PD-L1 expressed on tumor cells, PD-1 on immune cells, hampers function of i.e. T cells, maybe even just a cartoon.

Specific

Line 37 – has

Line 37 – “We explore a programmed”

Line 41 – “.. to establish a programmed”

Line 48 – suggest instead of suggested

Line 65 – a biomarker is generally something that can be measured in blood and/or tissue, perhaps another word would be better (or just markers)

Line 67 – to our knowledge can be left out as there is a literature reference to support this statement.

Line 71 – “… apoptosis, ferroptosis and autophagy”

Line 73 – comma can be removed

Line 135/136 – this line seems off.

Line 136 – please list all clinical variables, not just provide an example. If these were all the variables used, remove such as and restructure the sentence.

Author Response

Response to Reviewer 3

Response: Thank you for your comments.

(1) Programmed cell death is a way of cell death, and PD-1 is called Programmed Death-1. It is a very important immune checkpoint and a kind of protein on immune cells. The two concepts are different. (2) This paper mainly discusses the accuracy of programmed death prognostic model in predicting the prognosis of lung adenocarcinoma and its potential as a predictor of immunotherapy. The model is discussed as a whole and contains many prognostic genes. Therefore, it may be difficult to further study the mechanism of action, but important genes in it may be selected for further study in the future. (3) We improve the quality of our writing according to your suggestions.

We would like to thank you again for your valuable advice on our research.

Reviewer 4 Report

In this study, authors constructed a programmed cell death-associated gene prediction model to find that programmed cell death may affect TME and that anti-tumor-associated immune cells. However, several flaws should be revised before publication.

A flowchart or tables that represent the process (sample size, comparisons, analysis) should be included in detail to improve the reader's comprehension.

Authors should supplement more words in their study and the detailed parameters in the process of data analysis.

Please provide more details about the design scheme method and tell readers the difference and advantages of this model with others.

Authors should compare their results with available network methods or tools.

You can carry out a permutation test (Fisher exact test) that requires no distribution assumptions (require enough computational power). For multiple testing corrections, Benjamini-Hochberg is a widely used FDR algorithm and it has the best FP and FN controlling performance under independence assumptions.

In the "construction of networks and DEG analyses" part. Could the author describe in more detail (which package, analysis…) to build a network?

Data from different platforms need to be adjusted to remove any bias that may occur due to the difference in experiments/platforms. I would suggest using the TMM normalization method, ad the RNA-Seq data analysis pipeline used in the study is not very clear. The datasets were downloaded from different studies, authors need to adjust for the difference in batches, arrays, and other factors (like sample pooling, etc).

Principle component analysis (PCA) plots for showing the distance between samples need to validate the correctness of experimental design and DEGs.

For the sake of reproducibility, the author should release the code in the cloud or other repositories, such as GitHub, so that users can repeat it.

The figures are not inserted in the main manuscript context.

Brief definitions for "sensitivity" and "specificity" should be added.

Authors should perform further validation or follow-on biological studies to support the conclusions of the study. For example, qPCR was used to verify the differential expression of mRNA, etc.

 If possible, a table comparing age, gender, and possibly other demographics/comorbidities would be of interest. Are they matched? If not it is perhaps not possible to adjust for age/gender differences, but this should in this case be listed as another limitation as several of the putative pathways are likely affected by such differences.

Is there any correlation analysis between the download data?

In the material method, what data is used to construct the network?

In the method paper, a lot of analyses were carried out but not always were properly described.

More relevant and updated references should be added to the reference section.

Author Response

We appreciate your comments on our manuscript.

Based on your comments, we have made the following improvements to the manuscript:

  1. To improve the understanding of the readers, we have adjusted figure 1, adding the number of specific patients and the number of gene sets included in each cohort.

  1. In the methodology, we have clarified the design logic process of the article and added more information about the experiment and the algorithm. We have not further compare the performance of our model against other models since the data sets or processing methods are different, and simply comparing ROC curves will cause significant bias. However, in our model, the programmed death model composed of the ferroptosis model has a better predictive ability, due to its combination of apoptosis and autophagy.

  1. The main figures has been reinserted into the manuscript, which will make it easier for readers to understand

  1. In order to construct the difference analysis network, we conduct a spearman correlation test on pairs of DEGs, and we display the difference genes with correlation through a network diagram, which can be created using the R package “ggplot2”.

  1. Gene expression levels in 2 lung adenocarcinoma cell lines (A549, HCC827) were compared to those of normal bronchial epithelial cells (BEAS-2B) through quantitative PCR (qPCR).

  1. We are unable to obtain more detailed clinical information about patients due to the limitations of some data sets. However, our model research focuses on the overall survival rate of patients, and the ROC curve evaluation model shows that there is a high degree of predictive value.

  1. The patients we included came from different data sets, so we used the Combat algorithm to remove the batch effect for the RNA-seq data of all patients, and PCA analysis was used to determine whether samples were different before and after treatment. Our results confirm that the batch effect has been removed, and we can conduct further analysis.

Round 2

Reviewer 4 Report

The manuscript was improved.